# Residential Outdoor Environments for Individuals with Multiple Chemical Sensitivity (MCS)

**DOI:** 10.3390/ijerph22081243

**Published:** 2025-08-08

**Authors:** Emilia Danuta Lausen, Marina Bergen Jensen, Victoria Linn Lygum

**Affiliations:** 1Department of Geosciences and Natural Resource Management, University of Copenhagen, Rolighedsvej 23, 1958 Frederiksberg, Denmark; mbj@ign.ku.dk; 2Department of the Built Environment, Aalborg University, A. C. Meyers Vænge 15, 2450 Copenhagen, Denmark; vll@build.aau.dk

**Keywords:** environmental triggers, chemical sensitivity, housing units, design strategy, zoning, supportive outdoor environments, universal design

## Abstract

Severe sensitivity to various environmental chemicals affects an increasing number of people—a condition referred to as Multiple Chemical Sensitivity (MCS). The responses are both physical and psychological, where avoidance of chemical triggers can lead to social isolation, thereby increasing the level of disability. There is a need for user supportive environments where people with MCS can thrive, both indoors and outdoors. The study resulted in three principles for designing outdoor housing areas: (1) using spatial analysis to create a site layout that minimizes exposure to external and confounding triggers (e.g., noise, visual disturbances); (2) using zoning to clearly delineate private from semi-private areas; and (3) selecting vegetation and materials carefully to avoid triggers. The principles were developed in an iterative process based on existing research combined with a survey involving 58 MCS respondents.

## 1. Introduction

Multiple Chemical Sensitivity (MCS) is a complex medical condition characterized by severe sensitivity to various environmental chemicals in concentrations that do not affect most of the population [1]. Common triggers for individuals with MCS may include, among others, fragrances, cleaning products, pesticides, and vehicle emissions [2], while some individuals also are affected by some food items, metal, and electromagnetic waves [3].

The symptoms of MCS are varied, differ between individuals and may manifest as fibromyalgia, headaches, breathing difficulties, cough hypersensitivity, nausea, dizziness, and cognitive impairment [2,4]. Additionally, individuals affected by MCS may react to fungal spore allergens, such as mold, which can irritate their already sensitized mucous membranes (the Danish MCS Association’s website) and potentially lead to asthma attacks [5].

In addition to a wide range of physical triggers and responses, individuals with MCS may also experience psychological responses [1]. The mental strain from potential triggers, particularly odors beyond their control, such as those from neighbors, visitors, or activities like smoking, can result in anticipatory anxiety, which over time may develop into anxiety-related phobias [6]. Moreover, environmental stressors unrelated to chemical exposure, such as noise and electrical lighting, can further exacerbate MCS symptoms [6].

The prevalent strategy for managing symptoms among individuals with MCS is the avoidance of triggers, based on the condition’s physiological etiology. However, to date, no evidence-based protocols have been established [2]. This avoidance strategy involves the elimination of known chemical triggers, tailored to individual sensitivities, with the goal of creating a safe living environment [7,8]. Such approaches have been recommended following qualitative reviews of patients [9]. In the absence of established clinical treatments, individuals with MCS largely rely on self-management strategies to cope with their symptoms [2]. Unfortunately, trigger avoidance can lead to significant isolation from everyday activities, including work, social interactions, and public life, thereby increasing disability and reducing quality of life [10,11,12]. From this perspective, it is essential to design built environments that not only can alleviate MCS-related exposures but also foster social inclusion and well-being. Given the rising prevalence of MCS [13,14], addressing both the physiological and psychological dimensions of the condition within environmental design is increasingly important.

The focus on designing for a specific user group, in this case individuals with MCS, can be situated within the framework of Universal Design, an approach to architecture embracing diverse types of user needs in the built environment [15]. Universal Design can be described as “a process that increases usability, safety, health, and social participation through design and services that respond to the diversity of people and abilities” [16]. Its emphasis on health and protection from hazards [15] is particularly relevant when creating spaces for individuals with MCS. An essential first step in ensuring safe living conditions involves providing housing constructed from materials with low chemical emissions of chemical triggers that feature smooth, easily cleaned surfaces, eliminating conditions that promote mold growth, and integrating effective ventilation systems that deliver clean air while reducing outdoor air pollutants [17,18]. However, while some attention has been given to indoor environments, the outdoor surroundings of such residential units remain largely overlooked, with limited research exploring potential MCS triggers in exterior settings [19].

Contact with nature is known to significantly enhance human health by supporting physical, mental, and social well-being through exposure to natural elements such as greenery, greenspaces, and fresh air [20,21,22]. In this context, thoughtfully designed outdoor spaces within residential environments for individuals with MCS could foster nature connection, encourage social interaction, promote physical activity and potentially support immune function, contributing to a more holistic and health promoting living environment [20,21,22].

Furthermore, the ability to retreat from potential stressors is essential for individuals affected by MCS, both for physical and psychological reasons [23]. According to Gifford [24], access to secure, private spaces that allow personal control over environmental triggers and other stressors is vital for maintaining mental well-being and facilitating recovery from daily stress. For individuals with MCS, outdoor spaces should be designed to minimize exposure to environmental stressors while offering restorative, pleasant views and opportunities for safe social interaction to help mitigate the risk of isolation. Striking an effective balance between physical protection and maintaining visual access to, and connection with, the surrounding environment represents a critical design challenge.

The objective of this study is to develop design principles for outdoor areas at the residential unit level that support the needs of individuals with MCS, based on the current understanding of the condition.

## 2. Materials and Methods

### 2.1. Study Context

The study was part of a Danish project aiming to develop guidelines for constructing residential units tailored to individuals with MCS. It focused on key building considerations, including material selection, ventilation, room layout, and outdoor environmental factors. The team, composed of architects, engineers, and landscape architects, collaborated with a steering committee that included medical MCS specialists, indoor environmental chemists, and representatives from the Danish MCS Association (https://mcsforeningen.dk (accessed on 18 December 2024)). The latter provided input for the guidelines through a survey.

Lisbjerg, near Aarhus, Denmark, served as the case study site. Here, the housing organization Boligforeningen Ringgaarden plans to build approximately 50 rental units, designed as either terraced houses or apartments, specifically for people with MCS.

### 2.2. Study Approach

The study is grounded in the evidence-based design approach, defined as the “conscientious, explicit, and judicious use of current best evidence from research and practice in making critical decisions, together with an informed client, about the design of each individual and unique project” [25], p. 9.

Information about the MCS condition was gathered from experts involved in the project, including the project manager, project partners, and the experts in the steering committee. A major focus in the evidence-based design process is the inclusion of research performed by others, as well as answers to important questions found in new research [25]. In this context, individuals with MCS, who are prospective residents of the housing units, served as informed clients providing insights due to their direct experiential knowledge. This input was extracted from an online (SurveyMonkey) closed-question survey in Danish (Appendix A), completed by 58 MCS-affected individuals. The survey was conducted by *Lybech Landskab* with the involvement of all project partners. The survey link was sent to the members of the Danish MCS Association, and consent to use the survey findings was granted by all participants.

The proposed design principles and strategies were developed in an iterative process that incorporated various considerations, including the survey, and were visualized in simple sketch drawings. These drawings were later reviewed and commented on by the full project team and the steering committee before finalization.

The authors’ role in the process was to assist in the identification of key factors for outdoor environments, collecting and interpreting evidence, and developing design principles.

## 3. Results

### 3.1. Outdoor Areas’ Needs and Expectations

This section is based on the 58 responses from the SurveyMonkey questionnaire conducted among members of the Danish MCS Association.

#### 3.1.1. Outdoor Environmental Triggers


**Vegetation**


In response to the question, “What are your experiences with scented garden plants, such as lilac, lavender, and rose? (question 35, Appendix A)”, 62% of participants reported bad or additional 8% slightly bad experiences. In comparison, when asked “What are your experiences with spending time in gardens with flowering plants? (question 36, Appendix A)”, 44% of respondents indicated bad or slightly bad experiences. These results suggest a difference between the general experience of being in gardens with flowering plants and the experience of encountering specific, strongly scented plants. This may indicate that proximity to the source of the scent plays an important role in how it is perceived.


**Construction Materials**


Approximately 50% of the respondents reported poor experience with the smell of wet wood. Similarly, there were mixed experiences with wood as an indoor surface material, depending on the wood type and surface treatment. Untreated, degassed wood tended to cause fewer issues, but there was no consensus among respondents on which wood species are best tolerated. The risk of fungal spore development during wood decay was also mentioned as a factor of intolerance. The survey also indicated potential issues with the release of odors from wet concrete, while degassed, dry concrete was generally acceptable. Terraces with tile flooring were mentioned as problematic for some respondents, especially if the tiles were often wet or damp. However, problems with wet concrete and tile would be less pronounced with a covered terrace. As for the use of metal (iron, steel, and galvanized materials), there seemed to be no issues based on the survey responses.

#### 3.1.2. Outdoor Space Layout

In terms of the outdoor layout of residential units, 85% of the respondents expressed a strong desire for private outdoor space. Many respondents (35%) wished for a private terrace or balcony for fresh air, drying clothes (mentioned by 76% of respondents), placing items for off-gassing, or sleeping outdoors at night. An outdoor kitchen was considered unnecessary by 79% of respondents. Additionally, private outdoor areas needed to be well shielded from neighbors to prevent noise and the passing of odors. About half of the respondents saw it as advantageous if the private outdoor space could be easily covered by a roof, either fully or partially.

Regarding social interaction with the neighbors, only about 10% found it unimportant. In free-text responses, many mentioned the importance of having neighbors who respect the need for scent-free surroundings. When asked how many neighbors (including those with MCS), they would like to meet outdoors at a time, approximately 35% were comfortable with over 10 at a time, nearly 30% were undecided, and the rest were evenly distributed across 1–3, 4–5, and 6–10 neighbors at a time. Therefore, meeting places suitable for small groups as well as spaces designed for larger gatherings may be relevant. Around 75% would appreciate a shared drying yard for laundry, while a shared kitchen was not considered important. Shared workshops were mentioned as a desire in free-text responses.

Regarding the layout of outdoor spaces, the free-text responses consistently emphasized the importance of maintaining distance from sprayed fields, wood-burning stoves, and traffic.

In terms of the desired distance between housing and parking areas, approximately 20% of respondents answered 20–30 m, around 30% preferred 30–80 m, and about 25% preferred 80 to over 100 m. Therefore, it would be advantageous to have a portion of the housing, perhaps one-third, located relatively close to the parking area, while another third could be situated at a considerable distance, up to around 100 m.

### 3.2. Design Principles

The proposed design principles for MCS-friendly outdoor environments follow a trigger-avoidance strategy. Given the considerable variation in both the types of triggers and individual sensitivity thresholds, the principles focus on minimizing the presence and concentration of natural and synthetic airborne odors, as well as avoiding other confounding stressors such as noise and the unpredictable presence of others. There are two overarching principles applied at different scales: protective layers operating at the scale of the entire development area and avoidance through zoning operating at the scale of housing units. The considerations behind these two principles are presented in Table 1 and detailed below.

#### 3.2.1. Principle of Protective Layers

The protective layers’ principle aims to limit the number of triggers reaching the residential unit from the environment. A preliminary step is to consider the surroundings before purchasing the land for MCS residential units: The greater the distance to obvious sources of odors, air pollution and noise (e.g., industrial areas, heavily trafficked roads, conventionally cultivated fields with pesticides, manure, or pig farms), as well as other less apparent environmental factors such as residential areas with wood stoves, the better. Including these considerations from the start of the planning process will help limit the number of triggers reaching the residential unit.

Within the construction site, the protective layers are presented as a spatial model that divides the area into three main layers (Figure 1). The primary protective layer consists of vegetation, such as shrubbery planting with sparse trees. A 2–10 m high porous vegetation can serve as an effective wind break, capable of reducing pollen, odors, and noise to some degree [26,27,28,29].

The second sheltering layer is achieved by spatially organizing the buildings within the plot. By considering the prevalent wind directions and view directions to possible visual disturbances, the placing and sizing of hard structures can be utilized as sheltering elements. When positioning windbreaks and buildings, prioritizing sunlight is important.

The third layer is the resulting central green area, which benefits from the distance to residential units.

#### 3.2.2. Zoning Principles

The housing area, which is already protected by the primary boundary layer along the periphery, can be further divided into zones to facilitate a controlled and incremental exposure to the outdoor environment. By strategically planning these zones (Figure 2), the living environment can be tailored to the specific needs and preferences of individuals with MCS, balancing privacy, social interaction, and protection from MCS triggers (Table 1) (Figure 3, Figure 4 and Figure 5).

**Table 1 ijerph-22-01243-t001:** Description of placing, zone character and expected health benefits on the people with MCS per each zone.

	** *Zoning of the Area from Inside out* **			
	** *Private* **	** *Semi-Private* **	** *Semi-Public* **	** *Edge Planting (Protective Layer)* **
** *Placement* **	Directly attached or connected to a residence unit; can be accessed mainly by the inhabitant. IN TERRACED HOUSING An outdoor private zone established near the building as a terrace, either on the entrance side (entrance via terrace) or on the garden side (Figure 3A). IN APARTMENTS An outdoor private zone consisting of apartment balconies.	An access zone to residential entrances. IN TERRACED HOUSING Forms a semi-enclosed corridor connecting terraced houses’ entrances (Figure 4A). IN APARTMENTS Larger zone (could be multistorey) in front of an entrance to a residence or as a rooftop terrace (Figure 4B).	The remainder of the area that is placed within the property boundary (Figure 5). It is not entirely public due to sheltering (Figure 1) and the absence of flow-through traffic.	Zone along the outer boundary of the construction area.
** *Function and character* **	A personal, enclosed outdoor space. A scent-free and noise-free private spot, providing privacy from passers-by but with an outlook to semi-private zone. Provides an opportunity for inhabitants to decide about the finishing materials and vegetation (within the accepted range, see paragraph: Design Considerations). IN TERRACED HOUSING A private zone on the garden side of a terraced house could be divided by a shed on one side and a planted green screen on the other. With a garden gate (dotted line, Figure 3A), the private zone can be clearly marked while allowing visibility of those passing by. A pergola can be used to create a form of overhead screening. The terrace could also potentially be covered. IN APARTMENTS The balconies are fully screened from neighbors and create an outdoor apartment room (Figure 3B).	An area designated for social interaction and sharing communal facilities among neighbors. Parking options if necessary. Can accommodate the opportunity for cultivating vegetables in designated areas.	Meeting places can include a community house, workshop, greenhouse, and more. Transportation to and from the area, delivery of goods should be organized with consideration for the residents’ communal needs. Parking is available near the first residences (Figure 5), but not directly on the site; however, there is access for unloading/moving.	Vegetation-based sheltering layer A wide zone of minimum 5 meters planted with shrub vegetation and smaller trees. If the zone is wide enough, a small recreational path in the middle can be included. Placements of solid waste station should be considered in this zone (Figure 5). Scent-experience layer. In this zone the shrub species with low/minimal scent may be allowed (see paragraph: Design considerations).
** *Health benefits and supportive features for people with MCS* **	Scent-dependent on the user. Preferably not north-facing for daytime sun exposure benefit. Reduces stress from exposure to uncontrollable factors expected in outdoor spaces. Provides option for self-controlled contact with nature	Scent-free. Gives opportunity for social contact with closest neighbors that can be avoided if necessary. Provides options of hands-on nature contact (e.g., vegetable gardens).	Scent-free. Supports social contact opportunities with larger groups of inhabitants.	Opportunities to experience low volume scented vegetation. Nature recreation opportunities (walking, running).

### 3.3. Design Recommendations

To facilitate scent-free outdoor environments, this section presents recommendations for selection of vegetation and construction materials.

#### 3.3.1. Vegetation Selection

A wide range of commonly used plants cannot be recommended for the design of MCS-friendly residential areas due to their fragrance (considered as pleasant or unpleasant), allergenicity, or the ability to cause eczema upon contact. For the case site, a list of relevant species has been developed (Appendix B). A plant is only included on the list if no references to strong odors or fragrances have been found, meaning a strict criterion has been applied. Several of the excluded species may only have a faint scent or may only emit a fragrance if their needles and leaves are crushed. Therefore, some species may still be used, depending on the context and purpose.

The species are presented by their growth form (ground cover/creeper plants, perennials, shrubs, trees, and climbing plants) and listed with Latin and English names. The list is based on the species that are commonly used in Danish gardens and other green areas but are not necessarily native. It should be emphasized that the plants presented here have been selected by consulting various botanical works and plant retailers’ websites, which are not peer-reviewed sources. Furthermore, the list is not exhaustive, and errors are possible.

#### 3.3.2. Outdoor Materials Selection

Similar to vegetation, the main criterion for outdoor material selection is lack of scent. This involves considering year-round weather conditions and the eventual degeneration of materials, which can lead to the development of fungal spores (e.g., wood). Additionally, the production of the material and use of chemicals in the production process need to be considered. Therefore, the focus in materials selection should be on choosing natural materials with minimal chemical maintenance. Based on the survey and literature study, the following materials are recommended: (1) wire mesh for constructing vertical screening, mounted on wooden posts with ground spikes, (2) gabions (stone-filled) for stabilizing slopes and terrain transitions, (3) steel plates, such as Corten steel, for marking edges, (4) gravel with 100% mineral content for paths, roads, or parking areas, (5) cut natural stones, such as granite cobblestones or stone plates, for marking and reinforcing edges along gravel paths, establishing paths or squares, and (6) natural stones for marking non-parking areas, framing of communal spaces, or as outdoor furniture.

Wood is still recommended despite the possibility of fungal attacks, as this risk can be reduced by ensuring that the wood is not in contact with soil or other damp media and that it can dry after rain. It is unclear whether individuals with MCS can tolerate wood treated with borosilicate (water glass), but it is likely as it is an odorless mineral product.

## 4. Discussion

### 4.1. Experience of Outdoor Environments by Individuals with MCS

The survey among members of the Danish MCS Association revealed that individuals with MCS react to various outdoor air-borne odors, and the degree to which they are affected by the same trigger can vary. These findings are consistent with other health-oriented research focusing on reactions to chemical exposures [1,30].

The findings suggest that, although the majority of respondents were unable to tolerate strongly scented flower species, other flowering species were generally tolerated in outdoor environments. Another knowledge gap concerns sensitivity towards natural flower fragrances compared to synthetic forms such as oils and perfumes [31,32].

Chemicals from construction materials such as asphalt, concrete, and wood were considered more challenging than vegetation fragrances [33,34]. Similarly, the odor of wet wood was also found to be problematic. A recent large-scale population study conducted in the United States on chemicals that can induce intolerance did not include the smell of wet materials as a potential trigger [30]. However, the survey responses may still be supported by the two-phase mechanism of chemical intolerance induction: sensitization and triggering. During the sensitization phase, exposure to substances like wood preservatives may trigger initial symptoms of aversion. In the subsequent triggering phase, these symptoms can generalize to a broader range of wood-related odors [35], which may explain the negative responses of some respondents to wet wood smells in this study.

The survey revealed a conflict between respondents’ desire for private, sheltered outdoor spaces and their social needs. Private spaces may hinder the social interaction essential for reducing alienation and anxiety in individuals affected by MCS. This high need for private space could also relate to the fact that a high percentage of the MCS-affected individuals need to self-manage their symptoms and triggers [2,11]. In this context, it seems reasonable that it is easier for people with MCS to monitor and manage their condition in spaces with limited access for others. However, the overall health of the MCS-affected individuals is a result of multiple factors outside of health and living conditions and includes also social and income support [36].

### 4.2. The Role of Design Principles in the Well-Being of Individuals with MCS

The results from the survey confirm that the range of symptoms and triggers in MCS is person-dependent [2,35,37]. While some individuals with MCS experience a wide range of symptoms related to multiple triggers, others react intensively to only a small number of triggers [13]. Considering this, the described design strategy for the outdoor housing environments, even though it focuses on a specific health issue, cannot accommodate the full range of triggers that might affect future inhabitants of residential units. Therefore, the strategy centered on two aspects related to the inhabitants’ well-being that can be addressed by the outdoor areas: (1) improving the control of environmental MCS triggers and (2) enhancing social relations.

#### 4.2.1. Design for Physical Control of Environmental Triggers

In comparison to indoor conditions, outdoor concentrations of airborne chemicals are likely to be significantly lower [38]. However, reducing chemical emissions in indoor environments is easier to implement, as it is possible to control the sources—both through the selection of construction materials during the building process and their use during the maintenance phase. In outdoor conditions, the chemical triggers can be transported by wind over large distances. Therefore, the design and the implementation of the residential units for individuals with MCS requires more thorough site analysis than traditional residential units, especially regarding wind conditions around the year. This points to the relevance of the protective layers’ principle.

The presented design strategy introduces the concept of trigger control using physical barriers. Green shelterbelts, for example, can help reduce pollen concentrations by acting similarly to vegetation belts that filter particulate air pollution [39]. Additionally, building structures can serve as a secondary physical barrier, offering further protection by blocking particles that bypass the initial green layer from entering residential areas. However, there may be a trade-off between screening against odors and noise sources on the one hand, and on the other hand ventilating the area to remove odors, for example, from areas where waste is handled, and ensuring air circulation throughout the area. For this reason, waste disposal stations could potentially be placed at entrances and downstream in the prevailing wind directions.

#### 4.2.2. Enhancing Social Relations and Well-Being Through Semi-Spaces

The developed zoning principles are based on the concept of semi-spaces, i.e., transitional zones that combine the properties of the surrounding zones. For example, a semi-private zone allows flexible movement patterns to avoid social interactions when desired, while still affording opportunities for engaging when needed. The concept of semi-public spaces in urban design has been proposed by Gehl [40], and semi-public spaces have been noted as potentially crucial in creating social settings, enhancing the social interactions in housing areas [41] .

Furthermore, Park and Evans [41] refer to semi-public spaces as an important design aspect for reducing the environmental stressors in urban settings. Similarly, the lack of high-quality interaction spaces in multi-residential units has been linked to increased social stress [42].

Therefore, the zoning principles use the concept of semi-spaces to present more nested zones for social interaction that are divided based on the size and type of social interaction. Semi-private zones are reserved only for a small number of close-by neighbors while semi-public zones provide opportunities for interaction among a larger number of residents.

Universal Design principles prioritize sensory accessibility by promoting multisensory engagement and equitable use of space. While minimizing exposure to fragrance is crucial for managing MCS, it is also important to acknowledge the positive role scent can play in enhancing quality of life. Implementing protective zoning strategies in outdoor residential spaces can help strike a balance between necessary avoidance and over-isolation, supporting both well-being and social inclusion.

## 5. Conclusions

Given that MCS is a complex condition with an unclear etiology and no established treatment, there is a pressing need for broader research into harmful environmental chemicals, their health impacts, and strategies for mitigation to ensure safe, inclusive living environments.

This study presents, to the authors’ knowledge, the first set of outdoor design considerations for housing developments tailored to individuals affected by Multiple Chemical Sensitivity. The proposed principles aim to guide the creation of safe, non-toxic environments that reduce harmful odors and support well-being and nature contact, thereby optimizing the chances of individuals with MCS to have access to outdoor recreational spaces.

These guidelines may assist professionals in the construction and housing sectors designed for individuals with environmental sensitivities, as their applicability extends beyond those specifically affected by MCS. However, to ensure the long-term well-being of residents with MCS, it is recommended to develop clear regulations and a social contract among all inhabitants. Post-occupancy evaluations could provide valuable insights into the effectiveness of these guidelines and inform future research.

## Figures and Tables

**Figure 1 ijerph-22-01243-f001:**
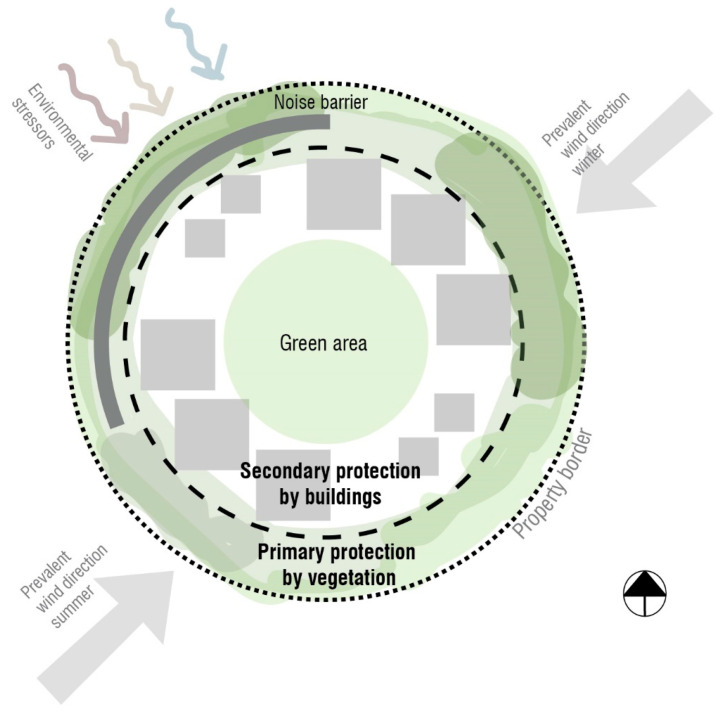
Illustration of the protective layers’ principle.

**Figure 2 ijerph-22-01243-f002:**
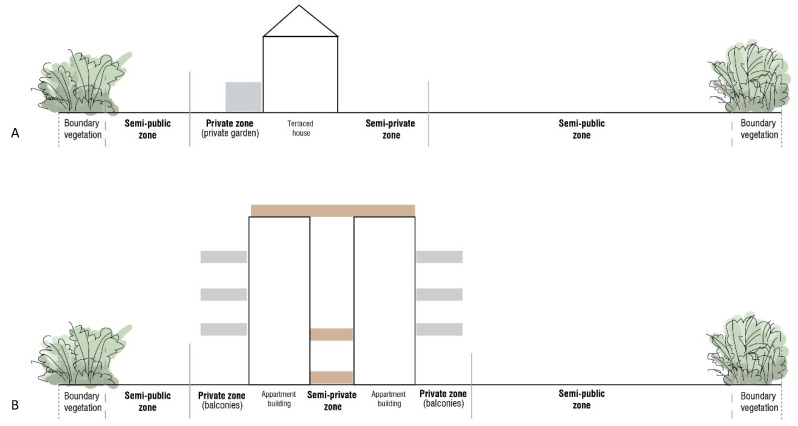
Division of zones across the plot for two types of residential units: (**A**) terraced housing; (**B**) apartments. In both cases, the private zone (grey) is near the residence either on the ground for terraced housing (**A**) or on the balcony or roof garden for apartments (**B**). The semi-public zones (neighbor zones, brown), located by the building entrances, are intended for arrival and passage, providing a transitional space between private and public areas. The boundary vegetation corresponds to the first protective layer, shielding against triggers from the external environment.

**Figure 3 ijerph-22-01243-f003:**
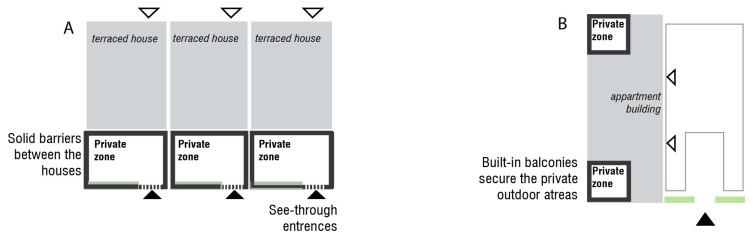
Allocation of the private zones in the terraced housing (**A**) and apartment housing (**B**). Arrowheads indicate building entrances.

**Figure 4 ijerph-22-01243-f004:**
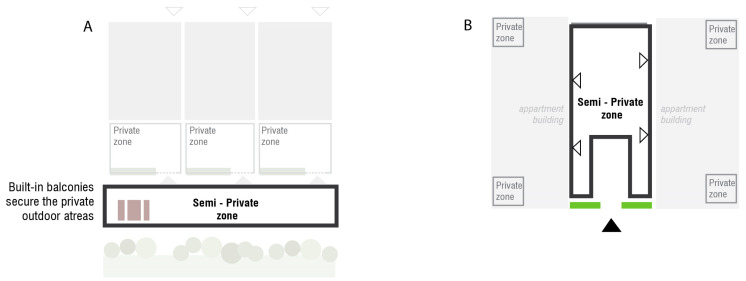
Allocation of the semi-private zones in terraced housing (**A**) and apartment buildings (**B**). Arrowheads indicate building entrances.

**Figure 5 ijerph-22-01243-f005:**
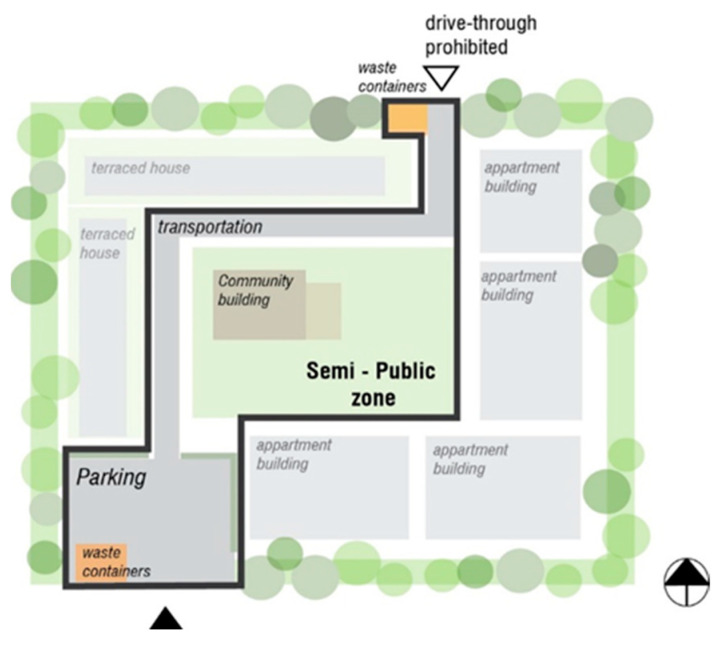
Allocation of the semi-public zone on the property. Parking should be restricted to one location within the property to minimize the car fumes in the housing area. Arrowheads indicate property entrances.

## Data Availability

The raw data supporting the conclusions of this article will be made available by the authors on request.

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
