# Peer review of "Residential Outdoor Environments for Individuals with Multiple Chemical Sensitivity (MCS)"

_ijerph, 2025, doi:10.3390/ijerph22081243_

Round 1
Reviewer 1 Report
Comments and Suggestions for Authors
Comments to the Authors:
- In general, it is ok and suitable for publication in your journal.
Abstract:
- Aim of manuscript clear in abstract, but results of study didn’t included.
Introduction:
- Although the introduction explained the meaning of the syndrome, its causes, and the purpose of the research, it did not address the results of previous research in terms of the concentration of stimuli that cause the problem, nor did it provide clear numbers for the concentrations at which the problem of the syndrome begins, to determine what amount of stimuli the person with the syndrome can tolerate and the permissible amount of exposure to it in order to provide a safe external environment for him. Authors should mention to results in the previous studies.
Materials and Methods
- Clear, ok.
Results
- Clear, but also did not contain clear list of various environmental chemicals affects people with MCS, like appendix B. The list should be contain ranges or numbers or percentage of environmental chemical and classify effects according to it such as acceptable, not acceptable, very sensitive, ....etc.
Discussion
- In lines 281-283: authors wrote “The survey among members of the Danish MCS Association revealed that individuals with MCS react to various outdoor air-borne chemicals, and the degree to which they are affected by the same trigger can vary”. Authors did not show various outdoor air-borne chemicals list, and the degree of effect?
- In lines 285-288: authors wrote “The findings indicate that while some respondents could not tolerate any flower fragrances, others were only sensitive to specific species”. Where this in result part?
- In line 320: authors wrote “airborne chemicals”. Authors didn’t show in results names or types or examples of airborne chemicals.
Conclusions
- In lines 320: authors wrote “The proposed principles prioritize the creation of safe, non-toxic environments that not only minimize exposure to harmful odors but also promote social well-being and nature contact”. Authors didn’t show in results any of that. Results did not have safe or minimum exposure dose of various environmental chemicals.
References
- ok
Reviewer 2 Report
Comments and Suggestions for Authors
Thank the authors of the paper for addressing the topic of multiple chemical sensitivity.
Comments to take into account:
- Review the citations on lines 41 and 76 in the Introduction; 106 and 111 in section 2.2.
- The authors use a survey to identify external disruptors, but they don't consider the possibility, or at least don't mention any questions, about exposure to electromagnetic waves, Wi-Fi, etc. as external triggers.
- Once sensitization occurs, the process appears to be irreversible and progressive. A myriad of substances can trigger a reaction, including cleaning products (80%); toiletries (shower gel, cosmetics, or perfumes: 75%); paints, varnishes, and solvents (50%); air fresheners, detergents, tobacco smoke, and fabric softener (20%); gasoline, tar, glue, and ink, among others (< 20%); solar exposure (29%), electromagnetic waves (10%); and sound waves or perceived seismic waves (< 6%). Some authors have postulated that the symptomatic trigger is not the substance itself, but the smell of it. Other stimuli comprise different foods and metals such as mercury. A relationship between sick building syndrome has been described in the literature. Please explain this aspect in a brief paragraph, as some patients are significantly affected by electromagnetic waves, WI-FI.
- Among the 58 patients who responded to the survey, include the female:male ratio, age, comorbidities, and known prior triggers that have led to avoidance behaviors, if recorded.
- Cite a recently published article in the Introduction, between lines 42 and 55. Navas-Soler, B., Palazó, A., Vallejo-Ortega, J., Santano-Pé, C., Seguí, C., Seguí, M., ... & Seguí, JM. (2025). Multiple Chemical Sensitivity: A Sickness of Suffering, Not of Dying. Descriptive Study of 33 Cases. Health, 17(1), 65-81. DOI: 10.4236/health.2025.171005
